# Fair Without Leveling Down: A New Intersectional Fairness Definition

**Gaurav Maheshwari, Aurélien Bellet, Pascal Denis, Mikaela Keller**
Univ. Lille, Inria, CNRS, Centrale Lille, UMR 9189 - CRIStAL, F-59000 Lille, France
`first_name.last_name@inria.fr`

## Abstract

In this work, we consider the problem of intersectional group fairness in the classification setting, where the objective is to learn discrimination-free models in the presence of several intersecting sensitive groups. First, we illustrate various shortcomings of existing fairness measures commonly used to capture intersectional fairness. Then, we propose a new definition called the $\alpha$-Intersectional Fairness, which combines the absolute and the relative performance across sensitive groups and can be seen as a generalization of the notion of differential fairness. We highlight several desirable properties of the proposed definition and analyze its relation to other fairness measures. Finally, we benchmark multiple popular in-processing fair machine learning approaches using our new fairness definition and show that they do not achieve any improvement over a simple baseline. Our results reveal that the increase in fairness measured by previous definitions hides a "leveling down" effect, i.e., degrading the best performance over groups rather than improving the worst one.

## 1 Introduction

The aim of fair machine learning is to develop models that are devoid of discriminatory behavior towards sensitive subgroups of the population. A broad range of approaches have been developed (see e.g., Zafar et al., 2017; Denis et al., 2021; Maheshwari and Perrot, 2022; Mehrabi et al., 2022, and references therein), with most of them focusing on a single sensitive axis (Lohaus et al., 2020; Agarwal et al., 2018) such as gender (e.g., Male vs. Female) or race (e.g., African-Americans vs. European-Americans). However, recent studies (Yang et al., 2020; Kirk et al., 2021) have demonstrated that even when fairness can be ensured at the level of each individual sensitive axis, significant unfairness can still exist at the intersection levels (e.g., Male European-Americans vs.

Female African-Americans). For example, Buolamwini and Gebru (2018) showed that commercially available face recognition tools exhibit significantly higher error rates for darker-skinned females than for lighter-skinned males. Similar observations have been made by several studies in NLP including contextual word representation (Tan and Celis, 2019), and generative models (Kirk et al., 2021). These findings resonate with the analytical framework of *intersectionality* (Crenshaw, 1989), which argues that systems of inequality based on various attributes (like gender and race) may "intersect" to create unique effects.

To capture these effects in the context of machine learning, several intersectional fairness measures have been proposed (Kearns et al., 2018; Hébert-Johnson et al., 2018; Foulds et al., 2020). Amongst them the most commonly used (Lalor et al., 2022; Zhao et al., 2022; Subramanian et al., 2021) is Differential Fairness (DF) (Foulds et al., 2020), which is the log-ratio of the best-performing group to the worst-performing group for a given performance measure (such as the True Positive Rate). While DF has many desirable properties, in this work we emphasize that DF implements a "strictly egalitarian" view, i.e., it only considers the *relative* performance between the group and ignores their *absolute* performance. In particular, a trivial way to improve fairness as measured by DF is by harming the best-off group without improving the worst-off group. This phenomenon, known as *leveling down*, does not fit the desired fairness requirements in many practical use-cases (Mittelstadt et al., 2023; Zietlow et al., 2022). Yet, we empirically observe that (i) popular fairness-promoting approaches tend to level down more in intersectional fairness, and (ii) this often goes unnoticed in the overall performance of the model due to the large number of groups induced by intersectional fairness.

To address these issues and explicitly capture the leveling down phenomena, we propose a gener-

alization of DF, called $\alpha$-**Intersectional Fairness** (IF$_\alpha$), which takes into account both the relative performance between the groups and the absolute performance of the groups. More precisely, IF$_\alpha$ is a weighted average between the relative and absolute performance of the groups, and allows the exploration of the whole trade-off between these two quantities by changing their relative importance via a weight $\alpha \in [0, 1]$. Our extensive benchmarks across various datasets show that many existing fairness-inducing methods aim for a different point in the aforementioned trade-off and generally show no consistent improvement over a simple unconstrained approach.

In summary, our primary contributions are as follows:

- We showcase the shortcomings of the existing intersectional fairness definition and propose a generalization called $\alpha$-**Intersectional Fairness**. We analyze the properties and behavior of the proposed fairness measure, and contrast them with DF.

- We benchmark existing fairness approaches on multiple datasets and evaluate their performance with several fairness measures, including ours. On the one hand, we find that many fairness approaches optimize for existing fairness measures by harming both the worst-off and best-off groups or only the best-off group. On the other hand, our measure is more careful in showing improvements over a simple baseline than previous metrics, allowing the emphasis on cases of leveling down.

## 2 Setting

In this section, we begin by introducing our notations and then formally define problem statement.

### 2.1 Notations

In this study, we adopt and extend the notations proposed by Morina et al. (2019). Let $p$ denote the number of distinct *sensitive axes* of interest, which generally correspond to socio-demographic features of a population. We refer to these sensitive axes as $A_1, \ldots, A_p$, each of which is a set of discrete-valued *sensitive attributes*. For instance, a dataset may be composed of gender, race, and age as the three sensitive axes, and each of these sensitive axes may be encoded by a set of sensitive attributes, such as gender: {male, female,

non-binary}, race: {European American, African American}, and age: {under 45, above 45}. We define a *sensitive group* $\mathbf{g}$ as any $p$-dimensional vector in the Cartesian product set $\mathcal{G} = A_1 \times \cdots \times A_p$ of these sensitive axes. A sensitive group $\mathbf{g} \in \mathcal{G}$ can then be written as $(a_1, \ldots, a_p)$ with $a_j \in A_j$.

### 2.2 Problem Statement

Consider a feature space $\mathcal{X}$, a finite discrete label space $\mathcal{Y}$, and a set $\mathcal{G}$ representing all possible intersections of $p$ sensitive axes as defined above. Let $\mathcal{D}$ be an unknown distribution over $\mathcal{X} \times \mathcal{Y} \times \mathcal{G}$ through which we sample i.i.d a finite dataset $\mathcal{T} = \{(x_i, y_i, \mathbf{g}_i)\}_{i=1}^n$ consisting of $n$ examples. This sample can be rewritten as $\mathcal{T} = \bigcup_{\mathbf{g} \in \mathcal{G}} \mathcal{T}_{\mathbf{g}}$ where $\mathcal{T}_{\mathbf{g}}$ represents the subset of examples from group $\mathbf{g}$. The goal of fair machine learning is then to learn an accurate model $h_\theta \in \mathcal{H}$, with learnable parameters $\theta \in \mathbb{R}^D$, such that $h_\theta : \mathcal{X} \to \mathcal{Y}$ is fair with respect to a given group fairness definition like Equal Opportunity (Hardt et al., 2016), Equal Odds (Hardt et al., 2016), Accuracy Parity (Zafar et al., 2017), etc.

Existing group fairness definitions generally consist of comparing a certain performance measure, such as True Positive Rate (TPR), False Positive Rate (FPR) or accuracy, across groups. In the following, for the sake of generality, we abstract away from the particular measure and denote by $m(h_\theta, \mathcal{T}_{\mathbf{g}}) \in [0, 1]$ the group-wise performance for model $h_\theta$ on the group of examples $\mathcal{T}_{\mathbf{g}}$, with the convention that **higher values of $m$ correspond to better performance**. For instance, in the case of TPR (used to define Equal Opportunity) we define $m(h_\theta, \mathcal{T}_{\mathbf{g}}) = \text{TPR}(h_\theta, \mathcal{T}_{\mathbf{g}})$, while for FPR, we define it as $m(h_\theta, \mathcal{T}_{\mathbf{g}}) = 1 - \text{FPR}(h_\theta, \mathcal{T}_{\mathbf{g}})$.

## 3 Existing Intersectional Framework

While the literature on group fairness in machine learning initially considered a single sensitive axis, several works have recently proposed fairness definitions for the intersectional setting (Gohar and Cheng, 2023). Kearns et al. (2018) proposed subgroup-fairness, which is based on the difference in performance of a particular group weighted by the size of the group. Several calibration and metric fairness-based variants were considered by Hébert-Johnson et al. (2018) and Yona and Rothblum (2018). A shortcoming of these notions is that they weight each group by its size, hence small groups may not be protected even though they are

often the disadvantaged ones.

## 3.1 Differential Fairness

To circumvent the above issue, Foulds et al. (2020) proposed Differential Fairness (DF), which puts a constraint on the relative performance between all pairs of groups. DF was originally proposed for statistical parity (Foulds et al., 2020), and was then extended by (Morina et al., 2019) to generalize other fairness definitions such as parity in False Positive Rates and Equal Odds. Below, we provide a general definition of DF based on an arbitrary group-wise performance measure $m$ as defined in Section 2.2.[1]

**Definition 1** (Differential Fairness). A model $h_\theta$ is $\epsilon$-differentially fair ($DF$) with respect to a group-wise performance measure $m$, if

$$\mathrm{DF}(h_\theta, m) \equiv \max_{\mathbf{g}, \mathbf{g}' \in \mathcal{G}} \ \log \frac{m(h_\theta, \mathcal{T}_{\mathbf{g}})}{m(h_\theta, \mathcal{T}_{\mathbf{g}'})} \leq \epsilon.$$

It is important to note that DF only depends on the relative performance between the best-performing group and the worst-performing group.

## 3.2 Shortcomings of Differential Fairness

We now highlight what we believe to be a key shortcoming of DF in the context of intersectional fairness: **DF can be improved by leveling down, i.e., harming the best-off and/or worst-off group, without significantly affecting the overall performance of the model.** This problem is caused by the combination of two factors.

First, DF is a strictly egalitarian measure that only considers the relative performance between groups. This can lead to situations where a model that improves the performance across all groups is deemed more unfair by DF. To illustrate this, let the group-wise performance measure $m$ to be the TPR and consider two models $h_\theta$ and $h_{\tilde{\theta}}$. Let the worst-off and best-off group-wise performance of $h_\theta$ be 0.50 and 0.60, respectively. For $h_{\tilde{\theta}}$, let it be 0.65 and 0.95. According to DF, $h_\theta$ is more fair than $h_{\tilde{\theta}}$ as the two groups are closer, while $h_{\tilde{\theta}}$ has better performance for both groups. In other words, $h_\theta$ is leveling down compared to $h_{\tilde{\theta}}$, but is deemed more fair. This exhibits the tension between the

relative performance between groups, and the absolute performance of the groups.

The second factor is that in intersectional fairness, leveling down can have a negligible effect on the overall performance of a model on the full dataset. This is because the number of groups in intersectional fairness is typically quite large (exponential in the number of sensitive axes $p$). Therefore, the bulk of examples generally do not belong to either the worst-off or best-off group, leading to a situation where the performance of other groups accounts for most of the model's overall performance. This issue may be further exacerbated if the class proportions are imbalanced across groups.

## 4  $\alpha$-Intersectional Fairness

In order to circumvent the above issue and effectively capture intersectional fairness while taking into account the leveling down phenomena, we propose $\alpha$-Intersectional Fairness (IF$_\alpha$). Our definition is essentially a convex combination of two components, namely (i) $\Delta_{rel}$, which takes into account the relative performance between the two groups, such as the ratio of their performance, and (ii) $\Delta_{abs}$, which captures the leveling down effect by accounting for the absolute performance of the worst-off group.

More precisely, given a model $h_\theta$ and a group-wise performance measure $m$, let us first define a measure of fairness for a pair of groups $\mathbf{g}$ and $\mathbf{g}'$:

$$I_\alpha(\mathbf{g}, \mathbf{g}', h_\theta, m) = \alpha \Delta_{abs} + (1 - \alpha) \Delta_{rel}, \quad (1)$$

where $\alpha \in [0, 1]$ and

$$\Delta_{abs} = \max \left( 1 - m(h_\theta, \mathcal{T}_{\mathbf{g}}), 1 - m(h_\theta, \mathcal{T}_{\mathbf{g}'}) \right),$$

$$\Delta_{rel} = \frac{1 - \max \left( m(h_\theta, \mathcal{T}_{\mathbf{g}}), m(h_\theta, \mathcal{T}_{\mathbf{g}'}) \right)}{1 - \min \left( m(h_\theta, \mathcal{T}_{\mathbf{g}}), m(h_\theta, \mathcal{T}_{\mathbf{g}'}) \right)}.$$

Now taking the maximum value of $I_\alpha$ over all pairs of groups, we get our proposed notion of $\alpha$-Intersectional Fairness.

**Definition 2** ($\alpha$-Intersectional Fairness). A model $h_\theta$ is $(\alpha, \gamma)$-intersectionally fair (IF$_\alpha$) with respect to a group-wise performance measure $m$, if

$$\mathrm{IF}_\alpha(h_\theta, m) \equiv \max_{\mathbf{g}, \mathbf{g}' \in \mathcal{G}} I_\alpha(\mathbf{g}, \mathbf{g}', h_\theta, m) \leq \gamma.$$

Note that IF$_\alpha(h_\theta, m)$ can be equivalently obtained as the the value of $I_\alpha$ over the pair of worst performing and the best performing group, as shown by the following proposition.

---

[1]We note that, in their extension to parity in False Positive Rates, Morina et al. (2019) did not account for the fact that higher FPR means lower performance, hence harming all groups always leads to better fairness. Our general formulation in Definition 1 fixes this problem through the convention that higher $m$ corresponds to better performance.

**Proposition 1.** If a model $h_\theta$ is $(\alpha, \gamma)$-intersectionally fair with respect to a group-wise performance measure $m$, then

$$\text{IF}_\alpha(h_\theta, m) = I_\alpha(\mathbf{g}^w, \mathbf{g}^b, h_\theta, m) \leq \gamma,$$

where $\mathbf{g}^w = \arg\min_{\mathbf{g} \in \mathcal{G}} m(h_\theta, \mathcal{T}_\mathbf{g})$ and $\mathbf{g}^b = \arg\max_{\mathbf{g} \in \mathcal{G}} m(h_\theta, \mathcal{T}_\mathbf{g})$.

In the following, we compare and contrast our fairness definition with DF when evaluating the fairness of the two models. We then investigate various properties of our proposed definition and discuss the impact of $\alpha$. In the interest of space, we delegate our discussion around the design choices for $\Delta_{rel}$ and $\Delta_{abs}$ to Appendix A.

**Comparing DF and IF$_\alpha$.** The primary difference between DF and IF$_\alpha$ when comparing two models arises when one model adversely affects the worst-off group ($\Delta_{abs}$) more than the other, despite having better relative performance ($\Delta_{rel}$). In this case, DF would consistently consider one model more fair than the other, whereas IF$_\alpha$ enables the exploration of this tension by varying the relative importance of both criteria through $\alpha$.

We formally capture this intuition as follows. Consider two models $h_\theta$ and $h_{\tilde{\theta}}$. Let the value of the worst-off and the best-off group's performance for the model $h_\theta$ be $w$ and $b$, respectively. Similarly, for model $h_{\tilde{\theta}}$ let the worst and the best group's performance be $\tilde{w}$ and $\tilde{b}$, respectively. Without the loss of generality, $\tilde{w}$ and $\tilde{b}$ can be written as $\tilde{w} = w + x$ and $\tilde{b} = b + y$. Note that $x$ and $y$ can be either positive or negative as long as $\tilde{w} \leq \tilde{b}$. We visualize this setup in Figure 1. Based on this setup, we have following cases:

- $x \geq y \geq 0$: In this case, $h_\theta$ harms the worst-off group (absolute performance) more, and its relative performance is worse than $h_{\tilde{\theta}}$. In Figure 1, this corresponds to $\tilde{w} \geq w$ and the blue region is smaller than the red region. Here, IF$_\alpha(h_{\tilde{\theta}}, m) \leq$ IF$_\alpha(h_\theta, m) \, \forall \, \alpha \in [0, 1]$, and DF$(h_{\tilde{\theta}}, m) \leq$ DF$(h_\theta, m)$.

- $x \leq y \leq 0$: This is similar to the case above, but with $h_{\tilde{\theta}}$ harming the groups more than $h_\theta$. In Figure 1, this corresponds to $\tilde{w} \leq w$ and the blue region is larger than the red region. Here, IF$_\alpha(h_{\tilde{\theta}}, m) \leq$ IF$_\alpha(h_\theta, m) \, \forall \, \alpha \in [0, 1]$, and DF$(h_{\tilde{\theta}}, m) \leq$ DF$(h_\theta, m)$.

- All other cases: In this setting, one of the model has better $\Delta_{abs}$ performance, while

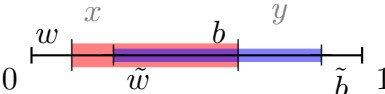

Figure 1: Group-wise performance range comparison. The range of group-wise performances of models $h_\theta$ and model $h_{\tilde{\theta}}$ are respectively $[w, b]$ and $[\tilde{w}, \tilde{b}]$. Note that the difference $x$ (resp. $y$) between the best (resp. worst) group-wise performances of $h_\theta$ and $h_{\tilde{\theta}}$ can be positive or negative.

the other model has better $\Delta_{rel}$ performance. The fairness in this setting depends on the relative importance of absolute and relative performance for IF$_\alpha$, while for DF it exclusively depends on absolute performance. In Figure 1, this corresponds to $\tilde{w} \leq w$ and the blue region is smaller than the red region or vice-versa. Here, $\exists \alpha \in [0, 1]$ for which IF$_\alpha(h_{\tilde{\theta}}, m) \geq$ IF$_\alpha(h_\theta, m)$ and vice versa. On the other hand, DF$(h_{\tilde{\theta}}, m) \leq$ DF$(h_\theta, m)$ if $y \times m(h_\theta, \mathcal{T}_{\mathbf{g^w}}) \leq x \times m(h_\theta, \mathcal{T}_{\mathbf{g^b}})$, otherwise DF$(h_{\tilde{\theta}}, m) >$ DF$(h_\theta, m)$.

To summarize, in the first two cases, one model harms the worst-off group (absolute performance), and the relative performance of that model is worse than the other. Thus, a good fairness measure should assign a higher unfairness to that model, which both DF and IF$_\alpha$ do. In the third case, one model performs better on the worst-off group, while the other model has a closer relative performance. The fairness in this setting depends on the relative importance of absolute and relative performance. Here, DF consistently assigns one model a higher fairness than the other, while IF$_\alpha$ enables to explore this tension and tune the relative importance of both criteria through $\alpha$. For instance, the previous example in Section 2 falls in the third case. On the one hand, DF will assign higher $\epsilon$ for $h_{\theta_1}$ in comparison to $h_{\theta_2}$. On the other hand, IF$_\alpha$ will assign higher $\gamma$ for $h_{\theta_1}$ for $\alpha \in (0.0, 0.81)$, while for all other $\alpha$, the $\gamma$ would be higher for $h_{\theta_2}$. We illustrate the effect of $\alpha$ in more details below.

**Impact of $\alpha$:** The parameter $\alpha$ allows to tune the relative importance of $\Delta_{abs}$ and $\Delta_{rel}$. On the one end of the spectrum, $\alpha = 0$ corresponds to considering only the relative performance $\Delta_{rel}$, while $\alpha = 1$ corresponds to considering only the absolute performance. At $\alpha = 0.0$ we recover the same relative ranking of unfairness as DF, and thus DF can be seen as a special case of IF$_\alpha$. In other words,

for any three models $h_{\theta_1}$, $h_{\theta_2}$, and $h_{\theta_3}$ such that $\text{DF}(h_{\theta_1}, m) \geq \text{DF}(h_{\theta_2}, m) \geq \text{DF}(h_{\theta_3}, m)$, then $\text{IF}_0(h_{\theta_1}, m) \geq \text{IF}_0(h_{\theta_2}, m) \geq \text{IF}_0(h_{\theta_3}, m)$. On the other end, $\alpha = 1$ only considers the absolute performance $\Delta_{abs}$), and $\alpha = 0.5$ corresponds to giving $\Delta_{abs}$ and $\Delta_{rel}$ an equal importance. In practice, it is useful to visualize the complete trade-off by plotting $\alpha \mapsto \text{IF}_\alpha$ (see Section 5).

**Intersectional Property:** We have the following intersectional property.

**Proposition 2.** Let the model $h_\theta$ be $(\alpha, \gamma)$-intersectionally fair over the set of groups defined by $\mathcal{G} = A_1 \times \cdots A_p$. Let $1 \leq s_1 \leq \cdots \leq s_k \leq p$, and $\mathcal{P} = A_{s_1} \times \cdots A_{s_k}$ be the Cartesian product of the sensitive axes where $s_j \in \mathbb{N}^+$. Then, $h_\theta$ is $(\alpha, \gamma)$-intersectionally fair over $\mathcal{P}$.

In other words, the fairness value calculated over the intersectional groups also holds over independent and "gerrymandering" intersectional groups (Yang et al., 2020). For instance, if a model is $(\alpha, \gamma)$-intersectionally fair in a space defined by gender, race, and age, then it is also $(\alpha, \gamma)$-intersectionally fair in the space defined by gender and race, or just gender. We delegate the proof to Appendix B.

**Generalization Guarantees:** $\alpha$-Intersectional Fairness enjoys the same generalization guarantees as the ones shown for DF in (Foulds et al., 2020). Indeed, the result of Foulds et al. (2020) relies on a generalization analysis of the group-wise performance measure $m$, which directly translates into generalization guarantees for $\text{IF}_\alpha$.

**Guidelines for setting $\alpha$:** $\alpha$-Intersectional Fairness enables exploring the tradeoff between worst-case performance and relative performance across groups. Indeed, at alpha=0.0, only relative performance is considered, aligning with strictly egalitarian measures. On the other extreme, at alpha=1.0, solely the worst-off group performance is considered. Based on this, we recommend:
Setting $\alpha = 0.75$ (more focus towards worst case performance) in:

- Situations where the cost of misclassification is not similar for each group. In these cases, leveling down would disproportionately affect those subgroups for whom the cost is higher. One example can be seen in education system, where the cost of denying financial assistance has higher impact on minority (Nora and Horvath, 1989; Hinojosa, 2023).

- Cases where data for disadvantaged groups is unreliable due to historical underrepresentation and lack of opportunities. For instance, certain facial recognition systems exhibit a higher likelihood of error when analyzing images of dark-skinned female individuals (Buolamwini and Gebru, 2018). Similarly, Sap et al. (2019) found that the hate speech detection systems are biased against black people.

In such contexts, emphasizing improvement for these disadvantaged groups is more pivotal than uniform performance over all subgroups, in line with the ideas of affirmative action. These scenarios best align with strategies seeking Demographic Parity or Equalized Odds.

Setting $\alpha = 0.25$ (more focus on relative performance) in:

- Scenarios where no group is significantly worse off, but to make sure that the algorithm behaves similarly for all the groups involved. This is related to algorithmic bias, as presented by Mehrabi et al. (2021). Moreover, the misclassification costs are similar in this setting.

- Legal or regulatory requirements may mandate similar outcomes across groups, like the 4/5th employment rule[2]. However, one must exercise caution when extrapolating this to other contexts, as it can lead to the "portability trap" as discussed by (Selbst et al., 2019).

In such a context, the emphasis is on equality among the groups. In practice, these scenarios indicate places where the partitioner would advocate for Accuracy Parity.

Otherwise, we recommend setting $\alpha = 0.50$ (a neutral default) when no domain or context-specific insights are available. This is what we used in our experiments. Ultimately, the choice of alpha reflects an understanding of the domain, the inherent biases in the data, and the real-world consequences of misclassifications.

---

[2]https://www.law.cornell.edu/cfr/text/29/1607.4

## 5 Experiments

In this section, we present experiments[3] that showcase (i) the model's performance over the worst-off group as the number of sensitive axes increases, and (ii) the "leveling down" phenomenon observed in various fairness-promoting mechanisms, along with the effectiveness of $\alpha$-Intersectional Fairness in uncovering it. However, before describing these experiments, we begin with an overview of the datasets, baselines, and fairness measures used.

**Datasets:** We benchmark over four datasets covering both text and images, with varying numbers of examples and sensitive groups:

- *Twitter Hate Speech*: The dataset is derived from multilingual Twitter Hate speech corpus (Huang et al., 2020) consisting of tweets annotated with 4 demographic factors (sensitive axes), namely age, race, gender, and country. The primary objective is to classify individual tweets as either hate speech or non-hate speech. In this work, we focus on the English subset and binarize all the demographic factors resulting in a total of 63 sensitive groups. Moreover, we only choose tweets where all the demographic factors are present. Consequently, our train, valid and test sets consists of $22,818$, $4,512$, and $5,032$ tweets.

- *CelebA* (Liu et al., 2015): The dataset consists of $202,599$ images of human faces, alongside 40 binary attributes for each image. We set 'sex', 'Young', 'Attractive', and 'Pale Skin' attributes as the sensitive axis for the images and 'Smiling' as the class label. We split the dataset into $80\%$ training and $20\%$ test split. Furthermore, we set aside $20\%$ of the training set as the validation split.

- *Psychometric dataset* (Abbasi et al., 2021): The dataset is a collection of $8,502$ free text responses alongside numerical scores over multiple psychometric dimensions. In this work, we focus on two dimensions:
  - *Numeracy* reflects the numerical comprehension capability of the individual.
  - *Anxiety* reflects the anxiety level as described by the patient.

Both these datasets consists of free text responses and binarized scores by the medical expert. Moreover, each response is associated with gender, race, age, and income. We use same pre-processing as (Lalor et al., 2022) and follow the same procedure to split the dataset as described above.

For improved readability, we present a subset of experiments in the main article. The remaining experiments are included in the Appendix.

**Methods.** We evaluate the fairness performance of the following methods: (i) `Unconstrained` which is oblivious to any fairness measure and solely optimizes the model's accuracy; (ii) `Adversarial` implements standard adversarial learning approach (Li et al., 2018), where an adversary is added to the `Unconstrained` with the objective to predict the sensitive attributes; (iii) `FairGrad` (Maheshwari and Perrot, 2022), is an in-processing approach that iteratively learns group-specific weights based on the fairness level of the model; (iv) `INLP` (Ravfogel et al., 2020), is a post-processing approach that iteratively trains a classifier to predict the sensitive attributes and then projects the representation on the classifier's null space. To enforce fairness across multiple sensitive axes in this work, we follow the extension proposed by Subramanian et al. (2021); (v) `Fair MixUp` (Chuang and Mroueh, 2021) is a data augmentation mechanism that enforces fairness by regularizing the model on the paths of interpolated samples between the sensitive groups.

In all our experiments, we employ the same model architecture for all the approaches to have a fair comparison. Specifically, we use a three-hidden layer fully connected neural network with 128, 64, and 32 corresponding sizes. Furthermore, we use ReLU as the activation with dropout fixed to $0.5$. We optimize cross-entropy loss in all cases with Adam (Kingma and Ba, 2015) as the optimizer using default parameters. Moreover, for Twitter Hate Speech and Numeracy datasets, we encode the text using bert-base-uncased (Devlin et al., 2019) text encoder. For CelebA, an image dataset, we employ ResNet18 (He et al., 2016) as the encoder. In all cases, we do not fine-tune the pre-trained encoders. Lastly, several previous studies have shown the effectiveness of equal sampling in improving fairness (Kamiran and Calders, 2009; Chawla et al., 2003; Kamiran and Calders, 2010; González-Zelaya et al., 2021). That is, to

---

[3]source code is available here: https://github.com/saist1993/BenchmarkingIntersectionalBias

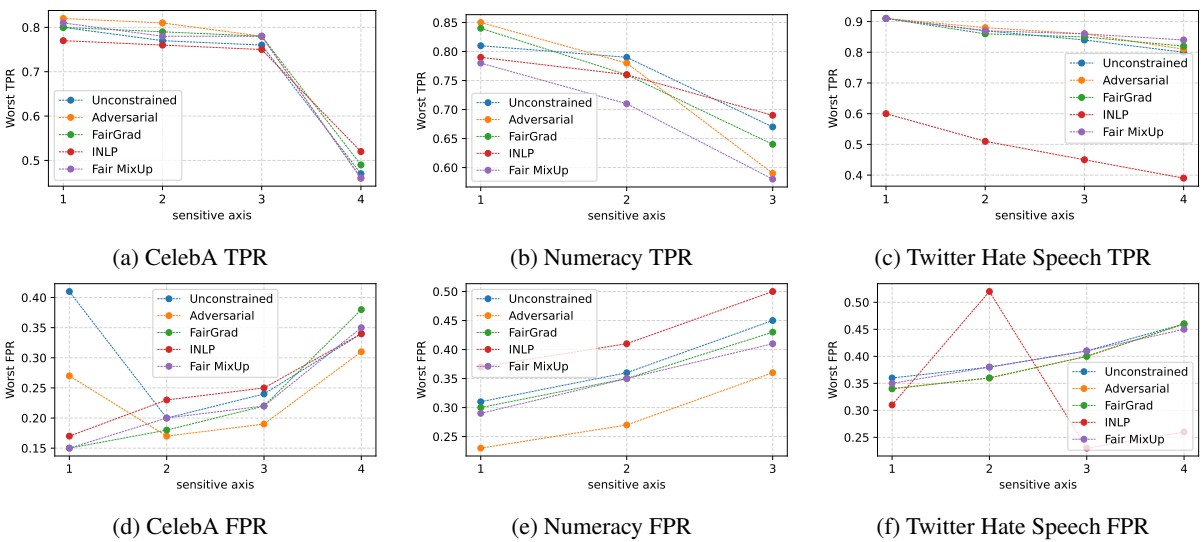

Figure 2: Test results over the worst-off group on *CelebA*, *Twitter Hate Speech*, and (b) *Numeracy* by varying the number of sensitive axes. For $p$ binary sensitive axis in the dataset, the total number of sensitive groups are $p^3 - 1$. Note that in FPR, lower the value better it is, while for TPR opposite is true.

counter the imbalance in the training data, the data is resampled so that there is an equal number of examples from each group and class in the final training set. Through preliminary experiments, we determine that equal sampling improves the worst-case performance of several approaches, including `Unconstrained` in various settings. We thus incorporate it as a hyperparameter indicating a continuous scale between undersampling and oversampling. Note that we also incorporate a setting where no equal sampling is performed, and we take the distribution as it is.

**Fairness performance measure.** In this work we focus on True Positive Rate parity and False Positive Rate parity as the fairness measure. The corresponding group wise performance measure $m$ for these fairness measures are TPR and FPR. Formally, $m$ in case of TPR for a group **g** is:

$$m(h_\theta, \mathcal{T}_\mathbf{g}) = P(h_\theta(x) = 1 | y = 1) \, \forall x, y \in \mathcal{T}_\mathbf{g},$$

while the FPR for a group **g** is:

$$m(h_\theta, \mathcal{T}_\mathbf{g}) = 1 - P(h_\theta(x) = 0 | y = 1) \, \forall x, y \in \mathcal{T}_\mathbf{g}$$

In order to estimate the empirical probabilities, we employ the bootstrap estimation procedure as proposed by Morina et al. (2019). In total, we generate 1000 datasets by sampling from the original dataset with replacement. We then estimate the probabilities on this dataset using smoothed empirical estimation mechanism and then average

the results over all the sampled datasets. In order to evaluate the utility of various methods, we employ balanced accuracy. Note that the choice of TPR Parity, and FPR Parity allows the derivation of several other fairness measures including Equal Opportunities, and Equalized Odds.

## 5.1 Worst-off performance and number of sensitive axis

In this experiment, we empirically evaluate the interplay between the number of sensitive groups and the harm towards the worst-off group. To this end, we iteratively increase the number of sensitive axes in the dataset and report the performance of the worst-off group for each approach. For instance, with CelebA we first randomly added gender (randomly chosen) when considering 1 sensitive axis. In the next iteration, we added race (randomly chosen) to the set with gender (previously added). Similarly, we then added age, and finally country. Note that for all the datasets, we start with a random choice of sensitive axis hoping to remove any form of selection bias. To select the optimal hyperparameters for this experiment, we follow the same procedure described in (Maheshwari et al., 2022) with the objective to select the hyperparameters with the best performance over the worst-off group.

We plot the results of this experiment in Figure 2. The results over the Anxiety dataset, which follow similar trend, can be found in the Appendix C. Based on these results, we observe that as the number of subgroups increases, the performance of the

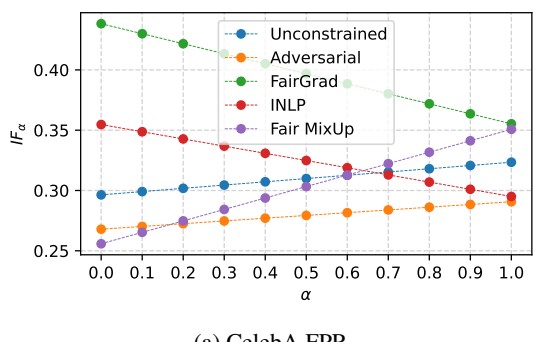

(a) CelebA FPR

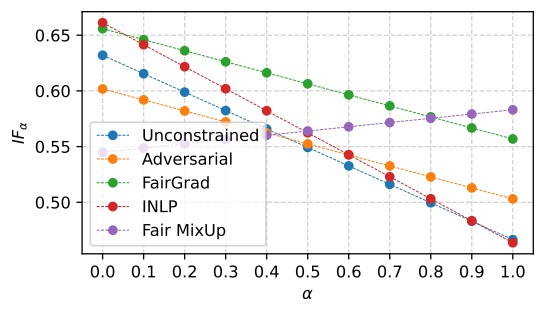

(b) Anxiety FPR

Figure 3: Value of $IF_\alpha$ on the test set of CelebA, and Numeracy datasets for varying $\alpha \in [0, 1]$.

| Method | BA ↑ | Best Off ↓ | Worst Off ↓ | DF ↓ | $IF_{\alpha=0.5}$ ↓ |
|---|---|---|---|---|---|
| Unconstrained | 0.81 + 0.0 | 0.08 + 0.01 | 0.36 + 0.04 | 0.36 +/- 0.06 | 0.31 +/- 0.02 |
| Adversarial | 0.8 + 0.0 | 0.07 + 0.02 | 0.32 + 0.02 | 0.31 +/- 0.12 | 0.28 +/- 0.04 |
| FairGrad | 0.77 + 0.01 | 0.14 + 0.01 | 0.39 + 0.01 | 0.34 +/- 0.03 | 0.4 +/- 0.02 |
| INLP | 0.8 + 0.0 | 0.09 + 0.01 | 0.34 + 0.04 | 0.32 +/- 0.03 | 0.32 +/- 0.01 |
| Fair MixUp | 0.8 + 0.0 | 0.08 + 0.01 | 0.37 + 0.02 | 0.38 +/- 0.04 | 0.3 +/- 0.01 |

(a) Results on CelebA

| Method | BA ↑ | Best Off ↓ | Worst Off ↓ | DF ↓ | $IF_{\alpha=0.5}$ ↓ |
|---|---|---|---|---|---|
| Unconstrained | 0.63 + 0.01 | 0.27 + 0.04 | 0.5 + 0.03 | 0.38 +/- 0.05 | 0.55 +/- 0.06 |
| Adversarial | 0.62 + 0.01 | 0.28 + 0.05 | 0.53 + 0.09 | 0.43 +/- 0.04 | 0.55 +/- 0.06 |
| FairGrad | 0.63 + 0.01 | 0.33 + 0.04 | 0.59 + 0.06 | 0.49 +/- 0.05 | 0.61 +/- 0.03 |
| INLP | 0.63 + 0.01 | 0.27 + 0.04 | 0.49 + 0.03 | 0.36 +/- 0.03 | 0.56 +/- 0.05 |
| Fair MixUp | 0.61 + 0.02 | 0.3 + 0.03 | 0.61 + 0.07 | 0.58 +/- 0.03 | 0.56 +/- 0.03 |

(b) Results on Anxiety

Table 1: Test results on (a) *CelebA*, and (b) *Anxiety* using False Positive Rate while optimizing for DF. The utility of various approaches is measured by balanced accuracy (BA), whereas fairness is measured by differential fairness DF and intersectional fairness $IF_{\alpha=0.5}$. For both fairness definition, lower is better, while for balanced accuracy, higher is better. The Best Off and Worst Off, in both cases lower is better, represents the min FPR and max FPR. Results have been averaged over 5 different runs. We deem a method to exhibit leveling down if its performance on either the worst-off or best-off group is inferior to the performance of an unconstrained model which we have highlighted using cyan ( ).

worst-off group becomes worse for all approaches in all settings. This can be attributed to the fact that the number of training examples available for each group decreases as the number of sensitive axis in the dataset increases. In terms of the performance of other approaches in comparison to Unconstrained, we find that fairness-inducing approaches generally perform better or similar to Unconstrained when 1 or 2 sensitive axes are considered. However, when 3 or more sensitive axis are considered, the performance of all approaches tends to converge to that of Unconstrained. For instance, in CelebA, on the one hand, with 1 sensi-

tive axis, all approaches significantly outperform Unconstrained with the difference between the best-performing method and Unconstrained being 0.26. On the other hand, when 4 sensitive axes are considered, the difference between the best-performing method and Unconstrained is 0.03, with only Adversarial outperforming it.

In a similar fashion, when considering TPR over Numeracy dataset, Unconstrained performs significantly worse than FairGrad and Adversarial with 1 sensitive axis while outperforming all approaches apart from INLP when 3 sensitive axis are considered. Similar observations can be made for

Numeracy and Twitter Hate Speech datasets in the FPR setting, with some minor exceptions. Overall we find that most fairness approaches start harming or do not improve the worst-off group as the number of sensitive axes grows in the dataset. Thus it is pivotal for an intersectional fairness measure to consider the harm induced by an approach while calculating its fairness.

## 5.2 Benchmarking Intersectional Fairness

In this experiment, we showcase the leveling down phenomena shown by various existing approaches. We also compare and contrast $IF_\alpha$ and DF. The results of this comparison over FPR parity can be found in Table 1a and 1b for CelebA and Anxiety respectively. The results of remaining two datasets over FPR parity, and all datasets over TPR Parity can be found in Appendix C. *In these experiment, we deem a method to exhibit leveling down if its performance on either the worst-off or best-off group is inferior to the performance of an unconstrained model.* In the results table, we highlight the methods that show leveling down in cyan (⬤).

We find that most of the methods have similar balanced accuracy across all the datasets, even if the fairness levels are different. This observation aligns with the arguments presented in Section 3 about the relationship between group fairness measure and the overall performance. In terms of fairness, most methods showcase leveling down. For instance, over the CelebA dataset, all methods apart from `Adversarial` shows leveling down. While in the case of Anxiety, all methods apart from `INLP` shows leveling down.

While comparing DF and $IF_{\alpha=0.5}$, we find that $IF_{\alpha=0.5}$ is more conservative in assigning fairness value, with most approaches performing similarly to Unconstrained. Moreover, leveling down cases may go unnoticed in DF. For instance, over the CelebA dataset, even though `FairGrad` and `INLP` showcases leveling down, the fairness value assigned by DF is lower for them than the one assigned to `Unconstrained`. Similar observation can be seen over Numeracy in case of `INLP`.

A particular advantage of $IF_\alpha$ over DF is that it equips the practitioner with a more nuanced view of the results. In Figure 3, we plot the complete trade-off between the relative and the absolute performance of groups by varying $\alpha$. For instance, in CelebA FPR, `Fair MixUp` shows the lowest level of unfairness at $\alpha = 0.0$. However, as soon as the worst-off group's performance is considered, i.e., $\alpha > 0.0$, it rapidly becomes unfair with it being one of the most unfair method at $\alpha = 1.0$. Interestingly, in Anxiety, `INLP` starts as one of the worst-performing mechanisms. However, with $\alpha > 0.0$, it quickly outperforms most approaches.

These findings shed light on the trade-offs and complexities inherent in optimizing fairness while maintaining worst-off group performance. It highlights the need for comprehensive evaluation metrics and the importance of considering the performance of both advantaged and disadvantaged groups in the fairness analysis. Finally, we emphasize that methods do not always exhibit leveling down. In settings without leveling down, DF adequately captures unfairness, producing values similar to $\alpha$-Intersectional Fairness. However, every method displays some degree of leveling down for some combinations of datasets and metrics. A robust fairness measure should expose unfairness universally, which our experiments demonstrate $IF_\alpha$ achieves.

## 6 Conclusion

We propose a new definition for measuring intersectional fairness of statistical models, in the group classification setting. We provide various comparative analyses of our proposed measure, and contrast it with existing ones. Through them, we show that our fairness definition can uncover various notions of harm, including notably, the leveling down phenomenon. We further show that many fairness-inducing methods show no significant improvement over a simple unconstrained approach. Through this work, we provide tools to the community to better uncover latent vectors of harm. Further, our findings chart a path for developing new fairness-inducing approaches which optimizes for fairness without harming the groups involved.

## 7 Acknowledgement

The authors would like to thank the Agence Nationale de la Recherche for funding this work under grant number ANR-19-CE23-0022, as well as the reviewers for their feedback and suggestions.

## 8 Limitations

While appealing, $\alpha$-Intersectional Fairness also has limitations. One of the primary ones is that it assumes a minimum number of examples for each subgroup to estimate the fairness level of the model

correctly. Moreover, it does not consider the data drift over time, as it assumes a static view of the problem. Thus we recommend checking the fairness level over time to account for it. Further, in this definition, setting up $\alpha$ is left to the practitioner and thus can be abused. In the future, we aim to develop mechanisms to validate $\alpha$ without access to the dataset or model.

Finally, we want to emphasize that a hypothetical perfectly fair model might not be devoid of social harm. Firstly, vectors of harm of using statistical models are not restricted to existing definitions of group fairness. Further, if some socio-economic groups are not present in a given dataset, existing fairness-inducing approaches are likely to not have any positive impact towards them when encountered upon deployment. Such is the case with commonly used datasets in the community, which over-simplify gender and race as binary features, ignoring people of mixed heritage, or non-binary gender, for example. In our experiments, we too have used these datasets, owing to their prevalence, and we urge the community to create dataset with non-binary attributes. That said, our measure works with non-binary sensitive attributes, with no modifications.

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

## A  Design Choices

In this section, we discuss our design choices for $\Delta_{abs}$ and $\Delta_{rel}$.

**Choice of $\Delta_{rel}$**  An alternate choice of $\Delta_{rel}$ is to utilize the performance difference between the groups instead of the above mentioned ratio. However, we advocate for the ratio as a superior choice for the following reasons:

- Scale-Invariant Comparison: The ratio enables comparing two models without the influence of the scale by normalizing the relative performance of a model. For instance, assume two models $h_\theta$ and $h_{\theta'}$ with the worst and the best group's performance for $h_\theta$ as $0.01$ and $0.02$ respectively, and $0.1$ and $0.2$ for $h_{\theta'}$. In this setting, the $\Delta_{rel}$ as the difference would always assign $h_\theta$ as fairer, even though both models are twice worse for the worst group compared to the best group. Note that our overall fairness measure accounts for the effect of scale through the inclusion of $\Delta_{abs}$. This is in-contrast to DFwhich does not take scale into account.

- Alignment with the $80\%$ rule: The ratio aligns with the well-known $80\%$ rule (Commission et al., 1990), which states that there exists legal evidence of discrimination if the ratio of the probabilities for a favorable outcome between the disadvantaged sensitive group and the advantaged sensitive group is less than $0.8$. By adopting the ratio as $\Delta_{rel}$, our metric adheres to this established criterion.

- Influence of worst-case group: If $\Delta_{rel}$ represents the difference in performance, then at $\alpha = 0.5$ the model with better worst-case performance will always have a lower $\gamma$ than the one with worse worst-case performance. In other words, at $\alpha = 0.5$, $\Delta_{abs}$ would always dominate $\Delta_{rel}$. However, this contradicts the intuitive understanding that, at $\alpha = 0.5$, both $\Delta_{rel}$ and $\Delta_{abs}$ should exert an equal influence.

**Choice of $\Delta_{abs}$**  An alternate choice we explored for $\Delta_{abs}$ was the average performance of the two groups involved instead of just the worst-performing one. However, Proposition 1 does not hold in the average case. This implies that a pair of groups can exist for which $I_\alpha$ is larger than the pair of groups consisting of the worst and best-performing groups. Moreover, Proposition 1 is an essential building block for intersectional property which is described later.

| Method | BA ↑ | Best Off ↑ | Worst Off ↑ | DF ↓ | IF$_{\alpha=0.5}$ ↓ |
|---|---|---|---|---|---|
| Unconstrained | 0.8 + 0.01 | 0.84 + 0.01 | 0.45 + 0.04 | 0.62 +/- 0.03 | 0.43 +/- 0.01 |
| Adversarial | 0.8 + 0.01 | 0.84 + 0.01 | 0.46 + 0.04 | 0.6 +/- 0.04 | 0.44 +/- 0.01 |
| FairGrad | 0.78 + 0.01 | 0.85 + 0.02 | 0.44 + 0.04 | 0.66 +/- 0.02 | 0.43 +/- 0.03 |
| INLP | 0.8 + 0.0 | 0.85 + 0.03 | 0.52 + 0.05 | 0.49 +/- 0.04 | 0.41 +/- 0.02 |
| Fair MixUp | 0.79 + 0.01 | 0.85 + 0.03 | 0.48 + 0.05 | 0.57 +/- 0.05 | 0.43 +/- 0.04 |

(a) Results on CelebA

| Method | BA ↑ | Best Off ↑ | Worst Off ↑ | DF ↓ | IF$_{\alpha=0.5}$ ↓ |
|---|---|---|---|---|---|
| Unconstrained | 0.68 + 0.02 | 0.87 + 0.05 | 0.61 + 0.04 | 0.36 +/- 0.01 | 0.38 +/- 0.09 |
| Adversarial | 0.7 + 0.01 | 0.81 + 0.05 | 0.55 + 0.08 | 0.39 +/- 0.03 | 0.45 +/- 0.07 |
| FairGrad | 0.68 + 0.02 | 0.88 + 0.04 | 0.64 + 0.09 | 0.32 +/- 0.03 | 0.35 +/- 0.07 |
| INLP | 0.68 + 0.01 | 0.84 + 0.05 | 0.66 + 0.1 | 0.24 +/- 0.03 | 0.44 +/- 0.08 |
| Fair MixUp | 0.7 + 0.01 | 0.81 + 0.05 | 0.54 + 0.05 | 0.41 +/- 0.02 | 0.44 +/- 0.07 |

(b) Results on Numeracy

| Method | BA ↑ | Best Off ↑ | Worst Off ↑ | DF ↓ | IF$_{\alpha=0.5}$ ↓ |
|---|---|---|---|---|---|
| Unconstrained | 0.79 + 0.01 | 0.96 + 0.01 | 0.77 + 0.03 | 0.22 +/- 0.03 | 0.2 +/- 0.03 |
| Adversarial | 0.76 + 0.0 | 0.97 + 0.01 | 0.81 + 0.04 | 0.18 +/- 0.04 | 0.21 +/- 0.03 |
| FairGrad | 0.76 + 0.02 | 0.95 + 0.01 | 0.78 + 0.03 | 0.2 +/- 0.04 | 0.25 +/- 0.03 |
| INLP | 0.67 + 0.01 | 0.73 + 0.03 | 0.38 + 0.03 | 0.65 +/- 0.05 | 0.56 +/- 0.03 |
| Fair MixUp | 0.76 + 0.01 | 0.98 + 0.0 | 0.84 + 0.02 | 0.15 +/- 0.02 | 0.16 +/- 0.01 |

(c) Results on Twitter Hate Speech

| Method | BA ↑ | Best Off ↑ | Worst Off ↑ | DF ↓ | IF$_{\alpha=0.5}$ ↓ |
|---|---|---|---|---|---|
| Unconstrained | 0.63 + 0.01 | 0.77 + 0.02 | 0.47 + 0.07 | 0.49 +/- 0.05 | 0.5 +/- 0.02 |
| Adversarial | 0.63 + 0.01 | 0.82 + 0.05 | 0.51 + 0.1 | 0.47 +/- 0.06 | 0.45 +/- 0.05 |
| FairGrad | 0.63 + 0.01 | 0.76 + 0.01 | 0.47 + 0.06 | 0.48 +/- 0.04 | 0.52 +/- 0.02 |
| INLP | 0.63 + 0.01 | 0.76 + 0.02 | 0.51 + 0.04 | 0.4 +/- 0.01 | 0.51 +/- 0.03 |
| Fair MixUp | 0.62 + 0.01 | 0.75 + 0.07 | 0.45 + 0.07 | 0.51 +/- 0.03 | 0.52 +/- 0.06 |

(d) Results on Anxiety

Table 2: Test results on (a) *CelebA*, (b) *Numeracy*, and (c) *Twitter Hate Speech* using True Positive Rate while optimizing for DF. The utility of various approaches is measured by balanced accuracy (BA), whereas fairness is measured by differential fairness DF and intersectional fairness IF$_{\alpha=0.5}$. For both fairness definition, lower is better, while for balanced accuracy, higher is better. The Best Off and Worst Off, in both cases higher is better, represents the min TPR and max TPR. Results have been averaged over 5 different runs. We have also highlighted methods which showcase leveling down using cyan ( ).

# B  Intersectional Property

In this section, we prove the intersectional property stated in Section 4. The proof follows the same procedure as described by Foulds et al. (2020). The intersectional property states that:

**Proposition.** Let the model $h_\theta$ be $(\alpha, \gamma)$-intersectionally fair over the set of groups defined by $\mathcal{G} = A_1 \times \cdots A_p$. Let $1 \leq s_1 \leq \cdots \leq s_k \leq p$,

and $\mathcal{P} = A_{s_1} \times \cdots A_{s_k}$ be the Cartesian product of the sensitive axes where $s_j \in \mathbb{N}^+$. Then, $h_\theta$ is $(\alpha, \gamma)$-intersectionally fair over $\mathcal{P}$.

The essential idea of the proof is to show that the maximum and the minimum group wise performance in $\mathcal{P}$ is bounded by the maximum and the minimum group wise performance in $\mathcal{G}$. After proving the above, then using Proposition 1, we

| Method | BA ↑ | Best Off ↓ | Worst Off ↓ | DF ↓ | IF$_{\alpha=0.5}$ ↓ |
|---|---|---|---|---|---|
| Unconstrained | 0.7 + 0.01 | 0.22 + 0.03 | 0.5 + 0.04 | 0.44 +/- 0.1 | 0.51 +/- 0.04 |
| Adversarial | 0.71 + 0.01 | 0.14 + 0.03 | 0.38 + 0.02 | 0.33 +/- 0.22 | 0.42 +/- 0.08 |
| FairGrad | 0.7 + 0.02 | 0.19 + 0.06 | 0.51 + 0.07 | 0.5 +/- 0.22 | 0.45 +/- 0.07 |
| INLP | 0.68 + 0.01 | 0.27 + 0.08 | 0.52 + 0.08 | 0.42 +/- 0.13 | 0.58 +/- 0.06 |
| Fair MixUp | 0.7 + 0.01 | 0.22 + 0.05 | 0.48 + 0.03 | 0.41 +/- 0.17 | 0.52 +/- 0.06 |

(a) Results on Numeracy

| Method | BA ↑ | Best Off ↓ | Worst Off ↓ | DF ↓ | IF$_{\alpha=0.5}$ ↓ |
|---|---|---|---|---|---|
| Unconstrained | 0.81 + 0.01 | 0.18 + 0.02 | 0.47 + 0.02 | 0.44 +/- 0.04 | 0.46 +/- 0.03 |
| Adversarial | 0.8 + 0.01 | 0.18 + 0.02 | 0.46 + 0.02 | 0.42 +/- 0.03 | 0.47 +/- 0.04 |
| FairGrad | 0.79 + 0.01 | 0.19 + 0.03 | 0.51 + 0.04 | 0.5 +/- 0.03 | 0.47 +/- 0.04 |
| INLP | 0.67 + 0.01 | 0.18 + 0.1 | 0.38 + 0.18 | 0.28 +/- 0.02 | 0.47 +/- 0.1 |
| Fair MixUp | 0.81 + 0.01 | 0.18 + 0.02 | 0.49 + 0.04 | 0.47 +/- 0.02 | 0.46 +/- 0.03 |

(b) Results on Twitter Hate Speech

Table 3: Test results on (a) *Numeracy*, and (b) *Twitter Hate Speech* using False Positive Rate while optimizing for DF. The utility of various approaches is measured by balanced accuracy (BA), whereas fairness is measured by differential fairness DF and intersectional fairness IF$_{\alpha=0.5}$. For both fairness definition, lower is better, while for balanced accuracy, higher is better. The Best Off and Worst Off, in both cases lower is better, represents the min FPR and max FPR. Results have been averaged over 5 different runs. We have also highlighted methods which showcase leveling down using cyan ( ).

can show that IF$_\alpha$ over $\mathcal{G}$ is higher than IF$_\alpha$ over $\mathcal{P}$.

Define $E = A_1 \times \ldots \times A_{a-1} \times A_{a+1} \ldots \times A_{k-1} \times A_{k+1} \times \ldots \times A_p$, the Cartesian product of the protected attributes included in $\mathcal{G}$ but not in $\mathcal{P}$. Then for any model $h_\theta$, $y \in \text{Range}(h_\theta)$,

$$\max_{\mathbf{g}\in\mathcal{P}:P(\mathbf{g}|\theta)>0} P_{h_\theta}(h_\theta(\mathbf{x}) = y|\mathcal{P} = \mathbf{g})$$

$$= \max_{\mathbf{g}\in\mathcal{P}:P(\mathbf{g}|\theta)>0} \sum_{\mathbf{e}\in E} P_{h_\theta}(h_\theta(\mathbf{x}) = y|E = \mathbf{e}, \mathbf{g})$$

$$P_{h_\theta}(E = e|\mathbf{g})$$

$$\leq \max_{\mathbf{g}\in\mathcal{P}:P(\mathbf{g}|\theta)>0} \sum_{\mathbf{e}\in E} \max_{\mathbf{e}'\in E:P_{h_\theta}(E=\mathbf{e}'|\mathbf{g})>0}$$

$$\left(P_{h_\theta}(h_\theta(\mathbf{x}) = y|E = \mathbf{e}', \mathbf{g})\right) \times P_\theta(E = \mathbf{e}|\mathbf{g})$$

$$= \max_{\mathbf{g}\in\mathcal{P}:P(\mathbf{g}|\theta)>0} \max_{\mathbf{e}'\in E:P_\theta(E=\mathbf{e}'|\mathbf{g},\theta)>0}$$

$$P_{h_\theta}(h_\theta(\mathbf{x}) = y|E = \mathbf{e}', \mathbf{g})$$

$$= \max_{\mathbf{s}'\in\mathcal{G}:P(\mathbf{s}'|\theta)>0} P_{M,\theta}(M(\mathbf{x}) = y|\mathbf{s}')$$

By a similar argument, $\min_{\mathbf{g}\in\mathcal{P}:P(\mathbf{g}|\theta)>0} P_{h_\theta}(h_\theta(\mathbf{x}) = y|\mathcal{P} = \mathbf{g}) \geq \min_{\mathbf{g}'\in\mathcal{G}:P(\mathbf{g}'|\theta)>0} P_{h_\theta}(h_\theta(\mathbf{x}) = y|\mathbf{g}')$. Applying Corollary 1, we hence bound $\gamma$ in $\mathcal{P}$ by the $\gamma$ in $\mathcal{G}$

## C  Extended Experiments

In this section, we detail the additional results. Table 2 provides results for the True Positive Rate (TPR) fairness measure, as outlined in the Experiment Section 5.2. In Figure 6, we vary the number of sensitive axes and plot the worst-case performance for Anxiety in FPR and TPR settings. Finally, Table 3 displays results related to the FPR parity fairness measure, focusing on the Twitter Hate Speech and Numeracy datasets. Notably, for TPR, each method exhibits leveling down in at least one dataset. For example, Adversarial shows leveling down in the Numeracy dataset, whereas INLP does so in both the Twitter Hate Speech and Anxiety datasets. Similarly, as with FPR, $DF$ does not consistently identify leveling down. As evidence, while both FairGrad and INLP demonstrate leveling down, they show a better fairness level than Unconstrained.

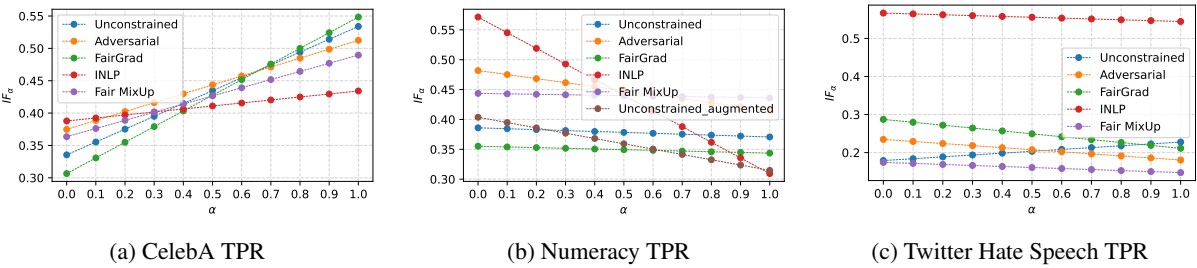

(a) CelebA TPR      (b) Numeracy TPR      (c) Twitter Hate Speech TPR

Figure 4: Value of $IF_\alpha$ on the test set of CelebA, Numeracy, and Twitter Hate Speech datasets for varying $\alpha \in [0, 1]$.

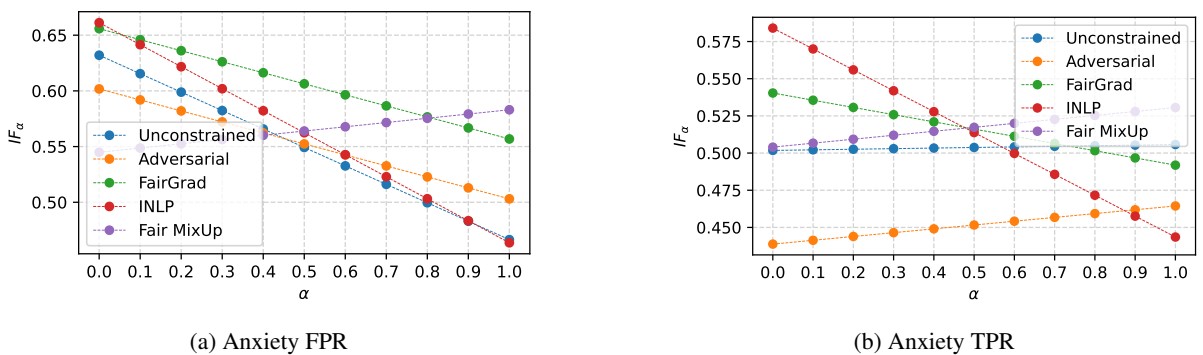

(a) Anxiety FPR      (b) Anxiety TPR

Figure 5: Value of $IF_\alpha$ on the test set of Anxiety datasets for varying $\alpha \in [0, 1]$.

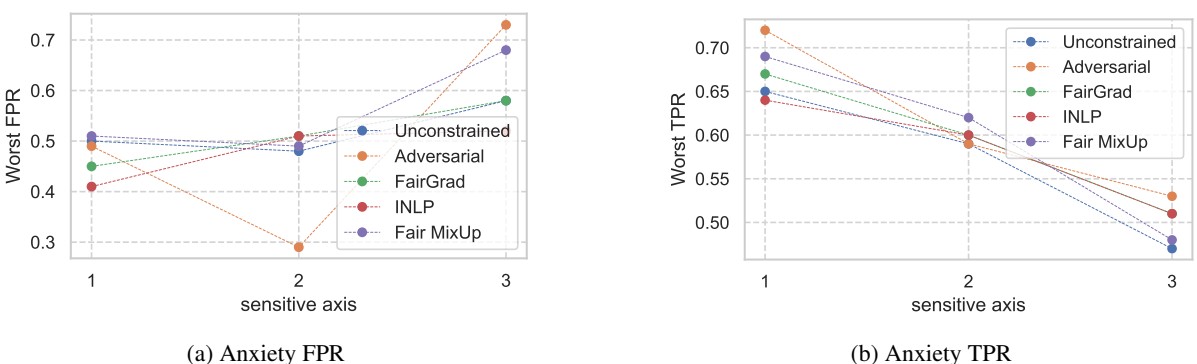

(a) Anxiety FPR      (b) Anxiety TPR

Figure 6: Test results over the worst-off group on *Anxiety* by varying the number of sensitive axes. For $p$ binary sensitive axis in the dataset, the total number of sensitive groups are $p^3 - 1$. Note that in FPR, lower the value better it is, while for TPR opposite is true.