# OpenReview forum: "Fair Without Leveling Down: A New Intersectional Fairness Definition"
_EMNLP/2023/Conference — EMNLP 2023 Main_

### Official Review · Reviewer_w78s · 2023-08-03

**Soundness:** 3

**Excitement:**

3: Ambivalent: It has merits (e.g., it reports state-of-the-art results, the idea is nice), but there are key weaknesses (e.g., it describes incremental work), and it can significantly benefit from another round of revision. However, I won't object to accepting it if my co-reviewers champion it.

**Paper Topic And Main Contributions:**

The researchers suggest that existing fairness measurements conceal a "leveling down" effect. Therefore, they propose a new measurement called $\alpha$-intersection fairness, which considers both absolute and relative performance across sensitive groups. They demonstrated their proposed measurement by benchmarking multiple popular in-processing fair machine learning approaches and observed that previous methods did not achieve improvement regarding this metric.

**Reasons To Accept:**

- The paper is well-written and has a clear structure. It is easy to understand the shortcomings of existing measurements and the reasons for proposing a new one.
- The proposed measurement is explained clearly.
- The claim about the leveling down effect is supported by experiments.
- Multiple datasets are used for cross-validation.

**Reasons To Reject:**

- The method of adding sensitive axes is not clearly explained. Did you add axes that belong to the same set, or were they added randomly?
- The result table does not show consistent improvement from other chosen metrics when compared to the baseline. How does this support the advantages of $\text{IF}_\alpha$?

**Reproducibility:**

4: Could mostly reproduce the results, but there may be some variation because of sample variance or minor variations in their interpretation of the protocol or method.

**Reviewer Confidence:**

2: Willing to defend my evaluation, but it is fairly likely that I missed some details, didn't understand some central points, or can't be sure about the novelty of the work.

---

> ### Author Rebuttal · Authors · 2023-08-28
>
> We would like to thank you for the review. Please let us know if we missed some of your points or if our comments and the other reviews raised additional questions.
>
>
> > The method of adding sensitive axes is not clearly explained. Did you add axes that belong to the same set, or were they added randomly?
>
>   We are not sure we understand the question exactly, but we do our best to clarify our setup below.
>
>   Each dataset we experimented with had pre-defined sensitive axes, either from the original dataset authors or commonly used in the community. For instance, the Twitter Hate Speech dataset had 4 binary sensitive axes - age with sensitive attributes as {U45, B45}, race with sensitive attributes as {African American, European American}, gender with sensitive attributes as {Male, Female}, and country with sensitive attributes as {USA, Outside USA}. This resulted in a total of 63 sensitive groups.
>
> - For the first experiment, we iteratively selected sensitive axes (and all the corresponding sensitive attributes) and added them to the set. For instance, with CelebA we first randomly added gender (randomly chosen) when considering 1 sensitive axis. In the next iteration, we added race (randomly chosen) to the set with gender (previously added). Similarly, we then added age, and finally country. Note that for all the datasets, we start with a random choice of sensitive axis hoping to remove any form of selection bias.
>
>  - For our second experiment we choose all the sensitive axes and their corresponding attributes.
>
>
> > The result table does not show consistent improvement from other chosen metrics when compared to the baseline. How does this support the advantages of IF-alpha?
>
> Please see our reply to the Reviewer, Msnu, about the improvements related to choosing $IF_{\alpha}$. In summary, $IF_{\alpha}$ does not improve performance as none of the models explicitly optimized for $IF_{\alpha}$. The primary objective of our experiments was to showcase leveling down phenomena and that $IF_{\alpha}$ is more resistant to it.

---

### Official Review · Reviewer_Msnu · 2023-08-05

**Soundness:** 4

**Excitement:**

3: Ambivalent: It has merits (e.g., it reports state-of-the-art results, the idea is nice), but there are key weaknesses (e.g., it describes incremental work), and it can significantly benefit from another round of revision. However, I won't object to accepting it if my co-reviewers champion it.

**Paper Topic And Main Contributions:**

This work is about the problem of intersectional group fairness in the classification setting. The main contribution is to propose a new intersectional fairness definition called $\alpha$-Intersectional Fairness. This new definition is mainly defined to balance the leveling down issue in old Differential Fairness.

**Reasons To Accept:**

This paper effectively demonstrate the harmful impact of the 'leveling down' issue, providing both a cogent explanation and compelling motivation for its new definition. They introduce a novel definition for intersectional fairness, termed '$\alpha$-Intersectional Fairness', whose definition is rigorous and well-articulated.

**Reasons To Reject:**

The work introduces another hyper-parameter $\alpha$. In the paper, it does not show how to decide this parameter $\alpha$. In the empirical experiment, there is not enough direct evidence to show how exactly the $\alpha$IF improves model performance.

**Reproducibility:**

5: Could easily reproduce the results.

**Reviewer Confidence:**

3: Pretty sure, but there's a chance I missed something. Although I have a good feel for this area in general, I did not carefully check the paper's details, e.g., the math, experimental design, or novelty.

---

> ### Author Rebuttal · Authors · 2023-08-28
>
> We would like to thank you for the review. Please let us know if we missed some of your points or if our comments and the other reviews raised additional questions.
>
> > Guideline for choosing alpha:
>
> Please see our reply to the Reviewer JvEV about guidelines for choosing alpha. We will add these guidelines to the final version of the paper.
>
> > How exactly the IF improves model performance:
>
> $IF_{\alpha}$ does not directly improve model performance since none of the evaluated methods explicitly optimize for $IF_{\alpha}$. We hope future work develops techniques tailored to optimizing $IF_{\alpha}$, which is out of the scope of this paper. The key takeaways of the current experiments are:
> - Experiment 1 shows existing methods can exhibit leveling down, harming involved groups. This is particularly evident in intersectional settings.
> - Experiment 2 demonstrates that current fairness measures are sometimes inadequate at uncovering the leveling down phenomena. In contrast, our $IF_{\alpha}$ is more conservative in measuring fairness, and most methods perform similarly to Unconstrained. This showcases the need for developing better fairness-inducing mechanisms. Moreover, $IF_{\alpha}$ helps explore tradeoffs between strictly egalitarian fairness views and improving worst-case performance (see Figure 3).
>
> Importantly, we emphasize that methods do not always exhibit leveling down. In settings without leveling down, DF adequately captures unfairness, producing values similar to $IF_{\alpha}$. However, every method displays some degree of leveling down for some combinations of datasets and metrics. A robust fairness measure should expose unfairness universally, which our experiments demonstrate $IF_{\alpha}$ achieves. In summary, no method displays leveling down in all cases, resulting in method's $IF_{\alpha}$ not being consistently worse than baseline (unconstrained) in every setting. Finally, following the suggestion of Reviewer JvEV, we will also add bar plots corresponding to Table 1, making this leveling down phenomena more explicit.

---

### Official Review · Reviewer_JvEV · 2023-08-05

**Soundness:** 4

**Ethical Concerns:**

Yes

**Excitement:**

3: Ambivalent: It has merits (e.g., it reports state-of-the-art results, the idea is nice), but there are key weaknesses (e.g., it describes incremental work), and it can significantly benefit from another round of revision. However, I won't object to accepting it if my co-reviewers champion it.

**Paper Topic And Main Contributions:**

The paper proposes a new intersectional fairness definition. The definition new definition is a linear combination of two components: (1) $\Delta_{ref}$, which takes into account the relative performance between the two groups,  and (2) $\Delta_{abs}$, which captures the leveling down effect by accounting for the absolute performance of the worst-off group. Through rigorous ad empirical analysis, they demonstrate the necessity of the proposed definition.

**Questions For The Authors:**

1. Could you please provide a concrete guideline for choosing $/alpha$ in the evaluation of intersectional fairness?

**Reasons To Accept:**

1. The paper is technically sound, and the proposed definition could provide a better evaluation and understanding of intersectional fairness.
2. The comparison of differential fairness is well-written, and the experiments are comprehensive.

**Reasons To Reject:**

1. The topic studied in this paper is less solely related to NLP. It is a general algorithmic fairness topic. The evaluation also uses CelebA – a vision dataset. I will suggest the author use Jigsaw dataset instead in the evaluation.

2. Choosing $\alpha$ for practitioners is hard. The author acknowledges the practitioner can abuse the choice of $/alpha$ in the limitation section. However, the author does not provide a guideline for choosing $\alpha$ in the evaluation, which should be critical.

3. Result visualization needs to be improved. For example, the results in Table 1 are hard to interpret. Maybe the bar charts could further improve the visualization by highlighting the difference between the best-offs and worst-offs.

**Reproducibility:**

4: Could mostly reproduce the results, but there may be some variation because of sample variance or minor variations in their interpretation of the protocol or method.

**Reviewer Confidence:**

3: Pretty sure, but there's a chance I missed something. Although I have a good feel for this area in general, I did not carefully check the paper's details, e.g., the math, experimental design, or novelty.

---

> ### Author Rebuttal · Authors · 2023-08-28
>
> We would like to thank you for the review. Please let us know if we missed some of your points or if our comments and the other reviews raised additional questions.
>
> > Guideline for choosing alpha:
>
> $\text{IF}_{\alpha}$ enables exploring the tradeoff between worst-case performance and relative performance across groups. Indeed, at alpha=0.0, only relative performance is considered, aligning with strictly egalitarian measures. On the other extreme, at alpha=1.0, solely the worst-off group performance is considered. Based on this, we recommend:
>
> - Setting $\alpha$=0.75 (more focus towards worst case performance) in:
> 	- Situations where the cost of misclassification is not similar for each group. In these cases, leveling down would disproportionately affect those subgroups for whom the cost is higher. One example can be seen in education system, where the cost of denying financial assistane has higher impact on moniroty [1][2].
> 	- Cases where data for disadvantaged groups is unreliable due to historical underrepresentation and lack of opportunities. For instance, certain facial recognition systems exhibit a higher likelihood of error when analyzing images of dark-skinned female individuals [3]. Similarly, [4] found that the hate speech detection systems are biased against black people.
>
> In such contexts, emphasizing improvement for these disadvantaged groups is more pivotal than uniform performance over all subgroups, in line with the ideas of affirmative action. These scenarios best align with strategies seeking Demographic Parity or Equalized Odds.
>
> - Setting $\alpha$=0.25 (more focus on relative performance) in:
> 	- Scenarios where no group is significantly worse off, but to make sure that the algorithm behaves similarly for all the groups involved. This is related to algorithmic bias, as presented by [5]. Moreover, the misclassification costs are similar in this setting.
> 	- Legal or regulatory requirements may mandate similar outcomes across groups, like the 4/5th employment rule [6]. However, one must exercise caution when extrapolating this to other contexts, as it can lead to the "portability trap" as discussed by [7].
>
> In such a context, the emphasis is on equality among the groups. In practice, these scenarios indicate places where the partitioner would advocate for Accuracy Parity.
>
> - Otherwise, we recommend setting $\alpha$=0.50 (a neutral default) when no domain or context-specific insights are available. This is what we used in our experiments.
>
>
> Ultimately, the choice of alpha reflects an understanding of the domain, the inherent biases in the data, and the real-world consequences of misclassifications.
>
> We agree with the reviewer that adding these discussions and guidelines will improve the quality and practical usefulness of our work. We will thus include them in the final version of the paper.
>
>
> > Result visualization needs to be improved:
>
> For the final version of the paper, we will use a bar chart alongside Table 1 to improve the readability.
>
> > Studied topic in this paper is less solely related to NLP:
>
> We included the CelebA dataset to demonstrate the broad applicability of our approach and show that the leveling down phenomenon generalizes beyond NLP contexts. Moreover, we believe intersectionality is pertinent to NLP, as it naturally arises in text data as multiple attributes of people are known to affect how they write [8][9].
>
> Thank you for suggesting the Jigsaw dataset. However, the identities in Jigsaw are not intersectional, which is the focus of this work. Indeed, each text in Jigsaw has a toxicity score and a single demographic identity such as feminist, queer, atheist, etc. Please let us know if we have misunderstood this suggestion. Instead of Jigsaw, we ran our experiments over the Anxiety dataset, which, akin to Numeracy, is part of the Psychometrics test set developed by [10]. The test consists of a free-text response about the patient's anxiety and a numerical score assigned by the medical expert. Each text is also associated with self-reported gender, race, age, and income. For pre-processing, we followed the footsteps of [11] and binarized each sensitive attribute. Please find a snippet of the results below (along the lines of Table 1). Our preliminary results reinforce our conclusion that methods showcase leveling down, and $\text{IF}_{\alpha}$ helps uncover it.
>
> | Method        | BA            | Best Off      | Worst Off     | DF            | $\text{IF}_{\alpha=0.5}$  |
> |---------------|---------------|---------------|---------------|---------------|---------------|
> | Unconstrained | 0.63 +/- 0.01 | 0.29 +/- 0.08 | 0.58 +/- 0.11 | 1.69 +/- 0.05 | 0.55 +/- 0.08 |
> | Adversarial   | 0.62 +/- 0.01 | 0.42 +/- 0.1  | 0.73 +/- 0.12 | 2.14 +/- 0.07 | 0.66 +/- 0.08 |
> | FairGrad      | 0.63 +/- 0.01 | 0.34 +/- 0.03 | 0.58 +/- 0.01 | 1.57 +/- 0.03 | 0.61 +/- 0.03 |
> | INLP          | 0.63 +/- 0.01 | 0.29 +/- 0.04 | 0.52 +/- 0.04 | 1.47 +/- 0.04 | 0.57 +/- 0.05 |
> | Fair MixUp    | 0.62 +/- 0.01 | 0.35 +/- 0.05 | 0.68 +/- 0.09 | 2.03 +/- 0.07 | 0.61 +/- 0.05 |
>
>
>
> Test results on Anxiety dataset using False Positive Rate while optimizing for DF. The utility of various approaches is measured by balanced accuracy (BA), whereas fairness is measured by differential fairness DF and intersectional fairness IFα=0.5. For both fairness definition, lower is better, while for balanced accuracy, higher is better. The Best Off and Worst Off, in both cases lower is better, represents the min FPR and max FPR. Results have been averaged over 5 different runs.
>
> | Method        | BA            | Best Off      | Worst Off     | DF            | $\text{IF}_{\alpha=0.5}$  |
> |---------------|---------------|---------------|---------------|---------------|---------------|
> | Unconstrained | 0.63 +/- 0.01 | 0.79 +/- 0.02 | 0.47 +/- 0.07 | 1.68 +/- 0.10 | 0.49 +/- 0.03 |
> | Adversarial   | 0.62 +/- 0.01 | 0.83 +/- 0.04 | 0.53 +/- 0.08 | 1.56 +/- 0.09 | 0.43 +/- 0.05 |
> | FairGrad      | 0.63 +/- 0.01 | 0.8 +/- 0.02  | 0.51 +/- 0.04 | 1.56 +/- 0.12 | 0.47 +/- 0.02 |
> | INLP          | 0.63 +/- 0.01 | 0.77 +/- 0.02 | 0.51 +/- 0.04 | 1.51 +/- 0.08 | 0.51 +/- 0.02 |
> | Fair MixUp    | 0.63 +/- 0.01 | 0.79 +/- 0.02 | 0.48 +/- 0.06 | 1.64 +/- 0.15 | 0.49 +/- 0.03 |
>
> Test results on Anxiety dataset using True Positive Rate while optimizing for DF. Rest everything remains the same as above.
>
>
> [1]: Nora, Amaury, and Fran Horvath. "Financial assistance: Minority enrollments and persistence." Education and Urban Society 21.3 (1989): 299-311.
>
> [2]:Hinojosa, Elisa Reyes. "Unequal Access to Higher Education: Student Loan Debt Disproportionately Impacts Minority Students." Scholar 25 (2023): 63.
>
> [3]: Buolamwini, Joy, and Timnit Gebru. "Gender shades: Intersectional accuracy disparities in commercial gender classification." Conference on fairness, accountability and transparency. PMLR, 2018.
>
> [4]: Sap, Maarten, et al. "The risk of racial bias in hate speech detection." Proceedings of the 57th annual meeting of the association for computational linguistics. 2019.
>
> [5]: Mehrabi, Ninareh, et al. "A survey on bias and fairness in machine learning." ACM computing surveys (CSUR) 54.6 (2021): 1-35.
>
> [6]: Law Cornell -  https://www.law.cornell.edu/cfr/text/29/1607.4
>
> [7]: Selbst, Andrew D., et al. "Fairness and abstraction in sociotechnical systems." Proceedings of the conference on fairness, accountability, and transparency. 2019.
>
> [8]: Peersman, Claudia, Walter Daelemans, and Leona Van Vaerenbergh. "Predicting age and gender in online social networks." Proceedings of the 3rd international workshop on Search and mining user-generated contents. 2011.
>
> [9]: Rangel, Francisco, et al. "Overview of the 3rd Author Profiling Task at PAN 2015." CLEF2015 Working Notes. Working Notes of CLEF 2015-Conference and Labs of the Evaluation forum.. Notebook Papers, 2015.
>
> [10]: Abbasi, Ahmed, et al. "Constructing a psychometric testbed for fair natural language processing." Proceedings of the 2021 Conference on Empirical Methods in Natural Language Processing. 2021.
>
> [11]: Lalor, John P., et al. "Benchmarking intersectional biases in NLP." Proceedings of the 2022 Conference of the North American Chapter of the Association for Computational Linguistics: Human Language Technologies. 2022.

---

### Meta-Review · Area_Chair_5FEE · 2023-09-18

**Recommendation:** 5

**Metareview:**

The paper proposes a new intersectional fairness definition that combines both absolute and relative performance across sensitive groups. The authors demonstrate the necessity of the proposed definition. The paper is well-written and has a clear structure. This technically sound paper could provide a better evaluation and understanding of intersectional fairness.

However, this paper lacks a description of how to choose $\alpha$. And there should be better evidence to show how exactly the
IF improves model performance. Including an NLP task is also recommended.

---

### Decision · Program_Chairs · 2023-10-07

**Decision:**

Accept-Main

**Comment:**

The paper proposes a new intersectional fairness definition that combines both absolute and relative performance across sensitive groups. The authors demonstrate the necessity of the proposed definition. The paper is well-written and has a clear structure. This technically sound paper could provide a better evaluation and understanding of intersectional fairness.

However, this paper lacks a description of how to choose $\alpha$. And there should be better evidence to show how exactly the
IF improves model performance. Including an NLP task is also recommended.